# Increased Risk of Fractures and Use of Proton Pump Inhibitors in Menopausal Women: A Systematic Review and Meta-Analysis

**DOI:** 10.3390/ijerph192013501

**Published:** 2022-10-19

**Authors:** Thuila Ferreira da Maia, Bruna Gafo de Camargo, Meire Ellen Pereira, Cláudia Sirlene de Oliveira, Izonete Cristina Guiloski

**Affiliations:** 1Faculdades Pequeno Príncipe, Curitiba 80230-020, Brazil; 2Instituto de Pesquisas Pelé Pequeno Príncipe, Curitiba 80250-060, Brazil

**Keywords:** fracture risk, menopause, proton pump inhibitors, woman

## Abstract

Proton pump inhibitors (PPIs) can directly interfere with osteoclastic function, induce hypergastrinemia, and inhibit calcium absorption, leading to reduced bone mineral density (BMD), a measure of bone metabolism that may be associated with the risk of fractures. The current study involves a systematic review and meta-analysis aimed at assessing the relationship between prolonged use of PPI drugs and fractures in menopausal women. A systematic search and meta-analysis were performed on PubMed, Scopus, and Science Direct databases according to PRISMA guidelines. Two independent reviewers analyzed the articles. The five articles found in the databases, which met the eligibility criteria, covered participants who were menopausal women aged between 56 and 78.5 years, using or not using a PPI for a minimum of 12 months. All studies showed an increase in the rate of fractures related to using PPIs, as an outcome. Prolonged use of PPIs in menopausal women can affect bone metabolism and cause fractures. However, other factors, such as the use of other classes of drugs, obesity, low weight, poor diet, replacement hormones, and comorbidities, should also be considered for assessing the risk of fractures.

## 1. Introduction

During menopause, an increase in bone resorption due to estrogen deficiency and a decrease in bone density is observed; this transition configures a potential stage for the development of osteoporosis and causes susceptibility to fractures at older ages. Menopause is a critical period for women in terms of changes in bone strength [1]. Therefore, this scenario can be considered an important risk factor for the occurrence of fractures, causing these conditions to alter the bone’s ability to resist trauma and influence bone geometry and its microarchitecture [1,2,3]. The fractures most likely to occur in this period are the vertebral, forearm, hip, and femur fractures [4].

In addition to osteoporosis, other risk factors may be associated with bone fragility, such as an association between the use of proton pump inhibitors (PPIs) and diseases that occur outside the upper gastrointestinal tract; in addition, there is growing concern regarding the possible relationship between use of these drugs and the risk for fractures [5,6,7]. PPIs were introduced in the late 1980s, and their use has considerably increased since [8]. These drugs act as acid suppressants and are used for treating diseases such as gastric and duodenal ulcers, gastroesophageal reflux, Zollinger Ellison syndrome, and in complementary therapy to eradicate *Helicobacter pylori* [9]. Its mechanism of action works by inhibiting the gastric H^+^/K^+^-ATPase enzyme, which is present in the cytoplasmic membranes of the parietal cell. Targeting multiple receptors to inhibit gastric acid secretion is the most effective therapy available for this purpose [10].

In a randomized crossover trial, O’Connell et al. [11] aimed to determine whether acid suppression caused by these PPIs would decrease calcium absorption in women. It concluded that omeprazole therapy significantly decreased calcium absorption from calcium carbonate when ingested by older women and that it could also occur with other PPIs through similar acid suppression mechanisms. Calcium absorption is the major determinant of calcium balance in the body, and its reduction has been directly identified as a cause of increased risk of osteoporotic fractures [8]. Khalili et al. [12] discussed a significant association between the regular use of PPIs and the risk of fracture, showing that risk increases with longer use of the drug, causing hip fracture among postmenopausal women.

Owing to the indiscriminate use of PPIs and serious adverse effects associated with their use, such as changes in bone metabolism, and knowing that menopausal women are at a greater risk of fractures, the objective of this systematic review was to assess whether there is a relationship between the use of proton pump inhibitors and fractures in menopausal women.

## 2. Materials and Methods

This systematic review and meta-analysis were conducted based on the standards established by the PRISMA declaration for reporting systematic reviews and meta-analyses of studies evaluating health interventions [13]. The review protocol was registered in PROSPERO and is available online (CRD42020218626).

To choose the guiding question, a PICO strategy was developed, where the acronym represents population (P), intervention (I), comparison (C), and outcomes (O). This resulted in the following strategy for constructing the study: (P)—menopausal women, (I)—take proton pump inhibitors, (C)—PPI users or non-users, (O)—are at risk for fractures. The resulting guiding question was: Can the use of proton pump inhibitors by menopausal women increase the risk of bone fractures?

### 2.1. Eligibility Criteria

The following inclusion criteria were used to select studies for analysis: (1) studies performed on humans, (2) studies with proton pump inhibitor drugs, and (3) studies performed on women in the menopausal phase. The exclusion criteria were as follows: (1) review articles, (2) studies that addressed the use of drug classes other than proton pump inhibitors, and (3) studies performed with men. The selected studies had no restrictions with respect to language or time periods.

### 2.2. Search Strategy

Search strategies for selecting the studies included using electronic databases, such as PubMed, Scopus, and Science Direct; using titles, abstracts, and keywords of the articles. A combination of the following keywords was used: woman, proton pump inhibitors, fracture risk, and menopause. The Boolean operator used to join these words was “AND”, resulting in the following combination: woman AND proton pump inhibitor AND fracture risk AND menopause. The articles from this search were transferred to Mendeley Desktop for reading titles and abstracts; subsequently, those meeting the eligibility criteria were selected. Two reviewers (TFM and BGC) independently assessed data from the articles. When there were discrepancies, they were resolved by consensus, or the opinion of a third reviewer was requested (ICG). The search of databases was carried out on 12 April 2021 and updated to date.

### 2.3. Data Extraction

Data were extracted from each article by two independent authors (TFM and BGC) and presented in a spreadsheet containing the following information: title, authors, journal, country where the study was conducted, study design, the total number of participants, age, period of menopause that was addressed, type of PPI used, period of PPI use, type of reported fracture, and the number of reported fractures. Random effects models were reported as risk ratios with 95% confidence intervals (CI). Other parameters that interfere with bone metabolism were also recorded when available, such as the mean age of study participants, other drug classes, use of vitamins, obesity, smoking, and hormonal therapy.

### 2.4. Study Quality Scale

The methodological quality of the studies was assessed using the Newcastle-Ottawa scale [14]. The methodological quality score of cohort studies was calculated using three components: group selection (0–4 points), comparability (0–2 points), and outcome (0–3 points). The maximum score was 9 points, which represents high methodological quality. Finally, each study was characterized as good, fair, or poor according to the number of stars obtained. The quality of a study was characterized as “good” when it obtained 3 or 4 stars in the selection domain, 1 or 2 stars in the comparability domain, and 2 or 3 stars in the outcome/exposure domain. “Fair” was used for 2 stars in the selection domain, 1 or 2 stars in the comparability domain, and 2 or 3 stars in the outcome/exposure domain. “Poor” was used in cases with 0–1 stars in the selection domain or 0 stars in the comparability domain, or 0–1 stars in the outcome/exposure domain. At this stage, two previously trained and qualified reviewers (TFM and BGC) independently assessed data from the articles; any discrepancies were resolved by consensus or by requesting the opinion of a third reviewer (ICG).

### 2.5. Meta-Analysis

Quantitative analyses were performed using PPI users and non-users as dichotomous variables. Data were weighted relative to the sample size of each study. Heterogeneity was assessed using the I-square index and ranked as follows: no heterogeneity (<25%), mild heterogeneity (25–50%), moderate heterogeneity (50–75%), and high heterogeneity (>75%) [15]. Additionally, funnel plots were used to detect the bias of the study. A meta-analysis was performed using a 95% confidence interval for the analyzed studies [12,16,17,18]. Dichotomous variables were analyzed using the Mantel-Haenszel method, and a random-effects model was applied. Results from the random effects models were reported as a risk ratio with a 95% confidence interval (CI). Statistical significance was set at *p* < 0.05. All analyses were performed using Review Manager© version 5.4.1 (Cochrane Collaboration, London, UK) and GraphPad Prism© version 8.4.2.

## 3. Results

### 3.1. Search Strategy

The search of databases resulted in 270 articles (Figure 1), of which 12 were obtained from PubMed, 16 from Scopus, and 242 from Science Direct. All 270 articles were transferred to Mendeley Desktop, and after removing duplicates (n = 11), 259 articles were selected for reading the titles and abstracts. After reading the titles and abstracts, 251 articles that did not fit the theme were excluded, leaving eight articles. Of these eight articles, one was excluded as a narrative review article, two did not meet the eligibility criteria, five [12,16,17,18,19] were selected for the development of the systematic review, and four [12,16,17,18] were selected for the meta-analysis. The study by Moberg et al. [19] could not be included in the meta-analysis because it only associated the occurrence of fractures with PPIs, describing the total number of participants and the total number of fractures.

### 3.2. Characteristics of Included Studies

After evaluating the studies, five met the eligibility criteria [12,16,17,18,19]. Of the five studies selected for the systematic review, two were conducted in the United States, one in Sweden, one in Germany and France, and one in Australia, all of which were observational prospective cohort studies published between 2009 and 2014.

Four articles covered a total of 9189 participants (menopausal women) in PPI use and 227,102 PPI non-users [12,16,17,18]. One of the studies, with 6416 women, did not specify how many participants were using or not using PPIs, but use was assessed as an increased risk factor for fractures. The women reported in these studies were in the post-menopausal phase and had a mean age between 56 and 78.5 years, part of the women were some type of PPI users, and this group was compared with the PPI non-user group. The use of PPIs reported in the studies was for a minimum of 12 months, and during the duration of the studies, these women could have had some type of fracture as a result of PPI use. The fractures described in these studies were hip (23.3%), spine (32%), vertebral (32.8%), forearm or wrist (26%), and total fractures (25%). These fractures were only described by Gray et al. [17]. The total number of fractures described was 811 for PPI users is 811 and 19,462 for PPI non-users. In addition, the study by Moberg et al. [19] described a total of 1137 fractures and associated these fractures with the use of PPIs (Table 1).

### 3.3. Quality Assessment of Studies

The overall methodological quality of the included studies was high (Appendix A). The Newcastle-Ottawa scale scores for these studies ranged from 8 to 9, with a rating higher than 8 indicating high quality. Both reviewers agreed on all occasions, and no study was excluded based on the risk of bias.

### 3.4. Relation of the Use of PPIs with Fractures

All the studies reported an increase in the occurrence of fractures related to PPI use (Appendix A).

Roux et al. [16] reported that patients using omeprazole had a history of mild fractures and a higher prevalence of vertebral fractures than the non-users. It also reported that the use of thiazides, thyroid hormones, and age are factors that can increase the risk of fractures but that the isolated use of omeprazole had already increased the risk. In multivariate analysis, omeprazole use was a significant and independent predictor of vertebral fractures (RR = 3.50; 95% CI 1.14–8.44).

Gray et al. [17] described that PPIs were not related to the risk of hip fracture but increased the risk of spine, forearm, or wrist fractures and the total number of fractures. For hip fractures, there was no evidence of effect modification in the subgroups examined. For all fractures, there was an interaction between PPI use and age. An increased risk of total fractures with the use of PPIs appeared in patients under 70 years of age. Among those with no history of fracture, the fracture risk was 32% higher in PPI users than in non-users. The interaction of drug classes was described as a risk factor for increased fractures, which may include psychoactive drugs for thyroid disorders, thiazide diuretics, coumarin, anticoagulants, loop diuretics, and blockers.

Khalili et al. [12] reported that the risk of hip fracture increased even in irregular users of PPIs. Even after stopping treatment with PPIs, these women had an increased risk of fractures for at least two years. After this period, they returned to the risk index for women who had never used PPIs. Compared to participants who did not regularly use PPIs, regular users had a higher body mass index, were less physically active, consumed less alcohol, were more likely to have osteoporosis, and were more likely to use hormone replacement therapy, thiazide diuretics and corticosteroids or bisphosphonates, which are factors that can increase the risk of fractures. The risk of increased fractures was also associated with smoking. In this study, there was no relationship between the age of the participants and the use of calcium replacement medication.

Lewis et al. [18], who linked the use of PPIs with the risk of fractures and falls, stated that long-term therapies were related to an increased risk of fractures, falls, and hospitalization. Falls that led to hospitalization were significantly higher in participants on long-term PPI therapy than in those who did not use PPIs. The risk of falls and fractures increased when PPIs were combined with corticosteroids and bisphosphonates.

Moberg et al. [19] reported that the use of PPIs alone or in combination with other factors increased the risk of fractures. Women with fractures were slightly taller and had a lower bone mass index (BMI) and bone density than women without fractures. Underweight women had an increased fracture risk both in gross terms and adjusted for age and smoking. In contrast, overweight women had a reduced fracture risk in the crude analysis but were not significant when adjusted for age and smoking. The duration of the fertile period was shorter in women with fractures than in those without fractures. A family history of diabetes was associated with reduced fracture risk, but diabetes itself had no impact on fracture risk. Increased fracture risk was observed in women using PPIs, with an odds ratio (OR) of 2.53 (95% confidence interval (CI)) 1.28–4.99.

All the studies mentioned that the participating women could use other drug classes, or vitamins, be overweight, use hormone therapy, or smoke, among other factors, which can contribute to changes in bone metabolism and alter the chances of increased fracture risk.

### 3.5. Meta-Analysis

The meta-analysis revealed that postmenopausal women who were PPI users had an increased risk of bone fractures (1.93 [1.39, 2.69]; *p* < 0.0001), and when the sensitivity analysis was performed, the outcome did not change, but the heterogeneity decreased from 92% to 15% (Figure 2).

## 4. Discussion

This systematic review and meta-analysis described the increased risk of fractures influenced by PPI use in menopausal women. Following the steps described in the search strategy, the five selected articles for the development of the study that met the eligibility criteria showed a relationship between the use of PPIs and the occurrence of fractures.

During menopause, serum estrogen levels can drop by up to 90%, resulting in greater bone resorption than bone formation [20]. The increase in bone resorption leads to accelerated bone loss and efflux of skeleton-derived calcium into extracellular fluid [21]. With the decline in estrogen levels during menopause, the rate of bone turnover increases, providing the main pathogenic mechanism for accelerated bone loss and the development of osteoporosis later in life [22]. These changes lead to the loss of total body calcium, resulting in skeletal loss [20]. Bone loss leads to deterioration of the skeletal microarchitecture and an increased risk of fracture [21]. In addition to menopause and osteoporosis, other factors may also be important for changes in bone metabolism.

PPIs can cause a decrease in bone reabsorption, led by inhibition of the osteoclast proton pump, generating greater chances of fracture, especially hip and vertebral fractures, in a population of men and women evaluated in the studies, without age restriction [23,24], This confirms the results obtained through meta-analysis performed in this study with menopausal women. According to Yang et al. [25], the duration of PPI therapy can influence the risk of hip fracture, especially when administered at high doses. A gradual increase in hip fracture was observed when exposure to PPI varied between 1–4 years of use. PPI type could also influence fracture risk, and a study conducted using rabeprazole showed the strongest association with fracture [26]. Another study developed by Poly et al. [27] showed an increased risk of hip fracture in 27% of those treated with rabeprazole and 13% of those treated with omeprazole and pantoprazole.

The physiological effect of therapeutic oral doses of omeprazole for gastric acid suppression on osteoclastic acidification remains unclear. Despite the high selectivity for H^+^/K^+^ ATPase, there is some activity of the drug in inhibiting the enzyme V-/H^+^ ATPase, which, based on scientific evidence, has already been associated with a reduction in bone resorption. The function of osteoclasts in bone resorption during bone growth and remodeling, which depend on the formation of an acidic extracellular compartment through the action of the vacuolar proton pump, requires continuous proton release. Therefore, the function of the V-/H^+^ ATPase enzyme is essential for these activities [8].

Based on the results of this study, the number of participants in the selected studies and time of PPI use were positive factors as they allowed for the assessment of the damage that PPIs can cause to the bodies of menopausal women. The number of fractures described in the studies, based on the duration of PPI use, also showed a good correlation between drug class and bone changes.

The main limitations of this study were the limited number of studies available for review and the confounding factors described by the authors. Most articles did not concretely conclude that PPIs interfere with bone metabolism due to the presence of other contributing factors.

## 5. Conclusions

Given that menopausal women already have a predisposition to fractures due to the occurrence of osteoporosis, it was observed that the use of PPIs could also affect bone metabolism and cause fractures; however, other individual issues of each PPI user should be evaluated for association with fractures. It is also necessary to assess the clinical condition for a drug prescription to avoid adverse effects. After performing this meta-analysis, it was observed that menopausal women who used PPIs had an increased relative risk for fracture of 1.93 (Figure 3).

## Figures and Tables

**Figure 1 ijerph-19-13501-f001:**
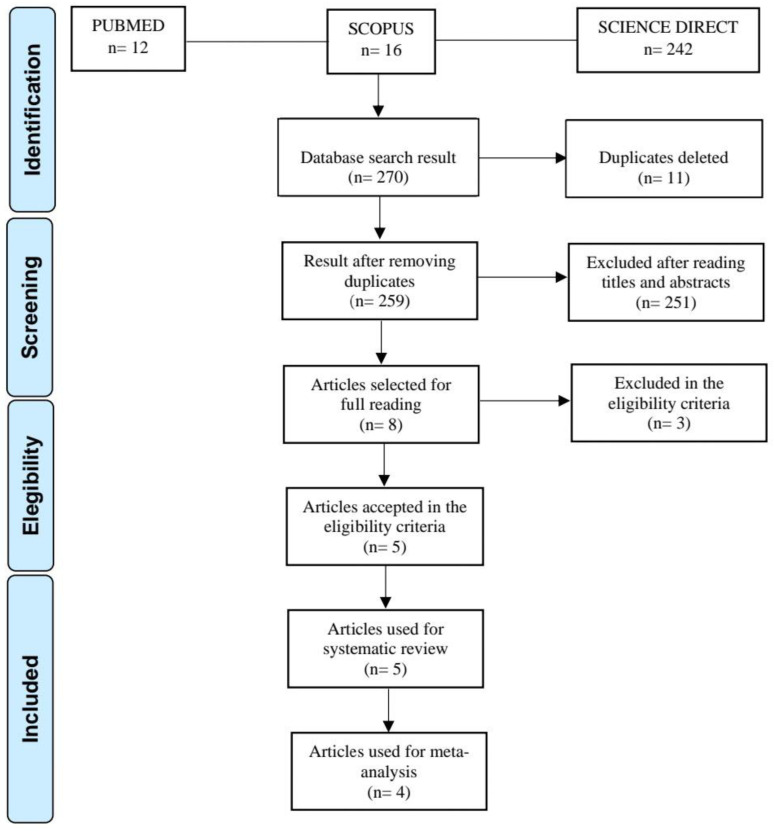
PRISMA flowchart demonstrating steps of the systematic review and meta-analysis.

**Figure 2 ijerph-19-13501-f002:**
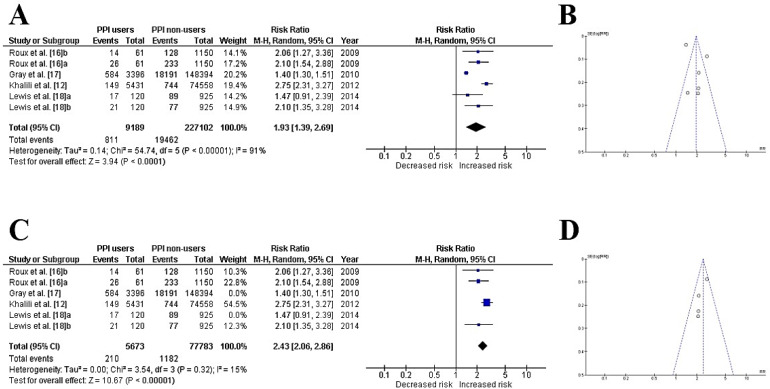
Forest plot for the standardized total of events among PPI users and non-users (**A**) crude forest and (**B**) funnel plots from all included studies; (**C**) forest and (**D**) funnel plots after trimming for heterogeneity and asymmetry.

**Figure 3 ijerph-19-13501-f003:**
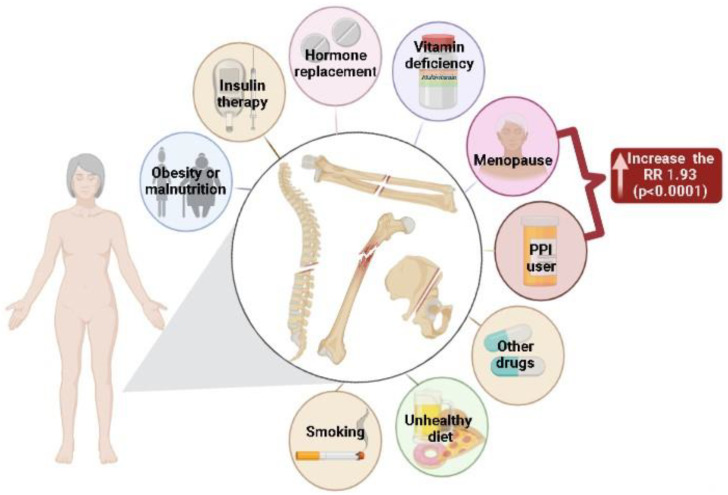
Risk factors for fractures in women.

**Table 1 ijerph-19-13501-t001:** Data from selected studies relating the fractures and the use of PPIs.

References	Year	Country	Kind of Study	Participants		Age (Years)	Fracture Sites	Fractures		PPI Use	PPI Use Duration
				PPI Users (n)	PPI Non-Users (n)			PPI Users (n)	PPI Non-Users (n)		
Roux et al. [16]	2009	Germany and France	Prospective cohort	61	1150	65.9 ± 6.5	Vertebral fractures	14	128	Omeprazole	12 months
				61	1150		Low-trauma fractures	26	233		
Gray et al. [17]	2010	United States	Prospective cohort	3396	148,394	64	Hip, spine, forearm, or wrist and total fractures	584	18,191	Lanzoprazole and omeprazole	12–36 months
Khalili et al. [12]	2012	United States	Prospective cohort	5431	74,558	66.8	Hip	149	744	*	24 months
Lewis et al. [18]	2014	Australia	Prospective cohort	120	925	78.5 ± 3.4	Hip—major fractures	17	89	Omeprazole, esomeprazole, pantoprazole, lansoprazole or rabeprazole	12 months
				120	925		Hip—fractures	21	77		
Moberg et al. [19]	2014	Sweden	Prospective cohort	6416 **		56.4	*	1137 **		*	*
Total				9189	227,102			811	19,462		

* Not reported by the author in his study; ** Described only the total number of participants and fractures.

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
