# Peer review of "Increased Risk of Fractures and Use of Proton Pump Inhibitors in Menopausal Women: A Systematic Review and Meta-Analysis"

_ijerph, 2022, doi:10.3390/ijerph192013501_

Round 1
Reviewer 1 Report
In the proposed systematic review includes 4-5 publications on PPI intake and a possible correlation to fracture risk. The topic is interesting and worth to highlight, however, there are several major comments I would like to address:
Introduction: The introduction, especially the background on PPIs and their association with bone disturbances is weak and does pinpoint to a possible interaction between osteoporosis and PPI intake. Moreover, there is a big break in Line 41 to 42. First, the authors describe the detrimental impact of menopause on bony outcomes and immediately jump to PPIs. There is no smooth bridge between this two topics.
Line 53-57: This sentence is very long and with repetitive words and meaning. Please rephrase.
Material and Methods:
Line 75-86: Which time frame did the authors use?
General issue: Since menopause is closely associated with an increased risk for osteoporosis, I am just wondering why the authors did not search for studies including “osteoporosis and PPI” or “osteoporosis, fracture and PPI”? This patient cohort is kind of forgotten in the systematic review without any reason. Please comment on this issue.
Results:
Line 131: The authors mention that the manuscripts included in the systematic review were published between 2009 and 2014. I am not convinced that there are no more new articles on that, however, if this is true, the topic may not be of interest anymore, maybe because some articles demonstrated that there is no effect. Please comment on this issue.
Line 140-141: There is a huge difference in PPI users and non-users with fractures. It is difficult to compare between such diverse groups. How can the authors overcome this issue?
Overall issue: the results’ section is very short and does not really highlight the main outcomes of the selected manuscripts. The authors just put everything together without discriminating the articles. I just recommend to include one paragraph per manuscript summarizing the main outcomes of each published article, which will make the review more readable.
Discussion:
The authors discuss a lot about vitamin intake etc. in the discussion part, without introducing anything in the introduction part. Please try to introduce the topics you will discuss later.
Conclusion:
Line 274: Please use the abbreviation PPI instead of “proton pump inhibitors”.
What about the patients with osteoporosis? The authors do not mention them at all in this part. Osteoporosis is the main cause of fractures in female elderly patients, therefore I kindly ask the authors not to forget on this important issue.
Author Response
Manuscript ID: ijerph-1900052
International Journal of Environmental Research and Public Health
Increased risk of fractures and use of proton pump inhibitors in menopausal women: a systematic review and meta-analysis
October 1st, 2022
Dear Editor Dr. Agnieszka Matuszewska,
Please find enclosed the corrections of the manuscript “Increased risk of fractures and use of proton pump inhibitors in menopausal women: a systematic review and meta-analysis ” by Thuila Ferreira Maia, Bruna Gafo Camargo, Meire Ellen Pereira, Cláudia Sirlene de Oliveira, and Izonete Cristina Guiloski and the answers to the editor and reviewers. We still would like this manuscript can be considered for publication in International Journal of Environmental Research and Public Health. Best wishes, and thanks again for the opportunity to resubmit the revised version.
The English were revised and the title changed.
Reviewer 1)
Open Review
English language and style
( ) I would not like to sign my review report
(x) I would like to sign my review report
( ) Extensive editing of English language and style required
( ) Moderate English changes required
(x) English language and style are fine/minor spell check required
( ) I don't feel qualified to judge about the English language and style
R: The English was revised and the title changed.
|
|
|
Comments and Suggestions for Authors
In the proposed systematic review includes 4-5 publications on PPI intake and a possible correlation to fracture risk. The topic is interesting and worth to highlight, however, there are several major comments I would like to address:
Introduction:
The introduction, especially the background on PPIs and their association with bone disturbances is weak and does pinpoint to a possible interaction between osteoporosis and PPI intake. Moreover, there is a big break in Line 41 to 42. First, the authors describe the detrimental impact of menopause on bony outcomes and immediately jump to PPIs. There is no smooth bridge between this two topics. Line 53-57: This sentence is very long and with repetitive words and meaning. Please rephrase.
R: The introduction has been rewritten relating topics bone disturbances, PPIs, and menopause. Lines 33-65.
Material and Methods:
Line 75-86: Which time frame did the authors use?
R: Lines 85-86 address that the languages or time of publication of the studies were not restricted.
General issue: Since menopause is closely associated with an increased risk for osteoporosis, I am just wondering why the authors did not search for studies including “osteoporosis and PPI” or “osteoporosis, fracture and PPI”? This patient cohort is kind of forgotten in the systematic review without any reason. Please comment on this issue.
R: The main objective of the systematic review was whether the use of PPIs by menopausal women would interfere with the increase in fractures, so the osteoporosis approach was succinct.
Results:
Line 131: The authors mention that the manuscripts included in the systematic review were published between 2009 and 2014. I am not convinced that there are no more new articles on that, however, if this is true, the topic may not be of interest anymore, maybe because some articles demonstrated that there is no effect. Please comment on this issue.
R: The initial search for the development of this systematic review was carried out in April 2021, and throughout the work, new searches were carried out to update the data, based on the chosen keywords. New articles were published, but none were relevant to the topic. Unlike other published systematic reviews, we want to emphasize that menopausal women are more vulnerable to fractures when using some type of PPI.
Line 140-141: There is a huge difference in PPI users and non-users with fractures. It is difficult to compare between such diverse groups. How can the authors overcome this issue?
R: It is common for the control to have a greater number of participants, but if we observe the relative risk for fracture in all articles selected in the same way, the relative risk is increased, and this assessment is important.
Overall issue: the results’ section is very short and does not really highlight the main outcomes of the selected manuscripts. The authors just put everything together without discriminating the articles. I just recommend to include one paragraph per manuscript summarizing the main outcomes of each published article, which will make the review more readable.
R: One paragraph per manuscript was inserted in the results summarizing the results found in the works. Lines 189-233.
Discussion:
The authors discuss a lot about vitamin intake etc. in the discussion part, without introducing anything in the introduction part. Please try to introduce the topics you will discuss later.
R: This was corrected and the discussion focused on the objectives of the study, in addition to being synthesized based on the suggestion of another reviewer.
Conclusion:
Line 274: Please use the abbreviation PPI instead of “proton pump inhibitors”.
R: It was done.
What about the patients with osteoporosis? The authors do not mention them at all in this part. Osteoporosis is the main cause of fractures in female elderly patients, therefore I kindly ask the authors not to forget on this important issue.
R: The conclusion was rewritten. Lines 292-298.

Reviewer 2 Report
This systematic review and meta-analysis is an interesting paper. The paper discusses the association between PPI use and future fractures in post-menopausal women. The strength of this paper is that it is novel regarding this specific topic and it addresses an important issue. It follows most of PRISMA’s guidelines for systematic reviews and meta-analyses. However, several items of the PRISMA 2020 checklist are not addressed https://prisma-statement.org/ (methods: item 10b, 12-13c, 13f-15 of the checklist, results: 16b-18, 20a, 20c-22 of the checklist and several sections of the results, discussion and conclusion lack clarity and depth as specified in the comments.
Major and minor comments
Major
- Although the authors state to have used the PRISMA guidelines for a systematic review many items are missing (e.g. . Please elaborate on specific topics in the methods:
a. The data that is extracted from the original articles. Why was it not only the primary outcome?
b. The date of data extraction should be added
c. Please elaborate on inclusion criteria for age, why do you choose such specific ages?
d. please state the elegibility criteria deriving from your PICO before the search strategy.
e. “Such as” line 101: is this an example of information that was extracted or a complete overview. Please give a complete overview as part of your methods according to the PRIMA guideline point 10a.
f. The authors do not state their assumption strategy in case of missing data.
g. paragraph 2.4: the threshold for high or low quality is missing
h. Information regarding the methods used in this meta-analysis is very limited. The metameter of interest (OR/HR etc.) is not named. The method for meta-analysis were not named.
i. Publication Bias was not assessed
j. possible sensitivity analysis was not stated.
- Results section
a. A clear overview and description of each study (and citation) is lacking. Please describe the main outcome measures of the studies (RR,HR?) in the text. Not only in the table.
b. Please, add references of included studies in the text of the results, as it is now unclear on which study you reflect (e.g line 129-131 but more examples could be given)
c. Line 163 -187: Please add effect sizes of included original research into the results
d. Line 160: Please give measures (RR 95%CI?) to interpret the numerical results in text
e. Please state in line 123 and 124 why one of the studies was not included in the meta-analysis and which one.
f. “All studies had as an outcome the increase in fracture index related to the use of PPIs” It is unclear what is meant by the fracture index
- Discussion section
a. Please review the references that are used. There are references are not suited for the purpose they are used for. E.g. line 45-48 states interference of PPI with calcium absorption and hypergastrenemia. However, they use a study on the relation between PPI and fractures as a reference (khalili etal). Khalili does not study the pharmacodynamics of PPI’s and their side-effects, they do mention the same mechanism in their introduction and reference this correctly. I would suggest referencing the original studies that do study the specifics on PPI use and possible mechanisms that could reduce BMD. Another example is the statement in line 33-34. “Their mechanism of action works by inhibiting the H+/K+-ATPase enzyme in gastric parietal cells, blocking acid secretion, and they are the most potent drugs available for this purpose”, while the main conclusion of the provided reference of Ali et al. is “Irrational prescribing of PPIs continues both in hospital and in general practice.”
b. The exposure of PPI use in this meta-analysis may be of special importance.
Since the duration of ppi use was 12 months in 2 out of 5 included studies, a bit longer (up to 36 months) in only 2 studies and in 1 study duration was not provided, it is not feasible to draw firm conclusions on ‘long term’ ppi use which is done in the discussion and conclusion. The finding that fracture risk is increased already in the first year of ppi use may point at confounding since a biological explanation for a rapid deterioration of bone quality resulting in an increased fracture risk in the first year of ppi use is not really available. This should be discussed.
Same for ppi dose, which is not addressed in the meta-analysis
c. The authors elaborate on the clinical characteristics that might add to or interfere with the association between PPI use and future fractures. But this is not an answer to the findings in the study related to the main research question. Also in the discussion the effect size they found in this meta-analysis is not compared to the existing body of literature.
- Although this is the first review on fracture risk in post-menopausal women. The article does not state clearly what the added benefit is in comparison to existing reviews and meta-analysis on the association between PPI use and Fractures.
Their aim is stated: “the objective of this systematic review was to assess whether there is a relationship between prolonged use of proton pump inhibitor drugs and fractures in menopausal women, as well as to assess whether the combination of these two factors can potentiate this risk.” The second part of the aim is not addressed in the results section. The answering of this question would require a comparison with risk measures from existing meta-analysis on the association between PPI use and fractures in the general population.
The title of the manuscript is not in line with conclusion of the study and therefore incorrect: the word ‘causes’ suggests a direct relationship between ppi use and increased fracture risk. The main conclusion of this meta-analysis however is that it is quite likely that the association between ppi use and fracture risk as found in this study is distorted by counfounding.
Minor
- Although I am not a native speaker, I have the impression that several sentences are not written in correct English.
- The structure of sentences with many commas makes it sometimes difficult and sometimes incorrect read. For example: line 26-40, line 53-57. line 192-200
- The rationale of some sentences is not clear:
line 135-138 Reported periods of menopause were post-menopause (this is not clear what you mean), aged 56 to 78.5 years (persons?), using some type of PPI and compared to the group that was not using (an PPI?), for a minimum period of 12 months (the use or the non-use) and that during the study had some type of fracture resulting from this use.
2. Please note: The minimum required word count for reviews of this paper is 4000 words.
Author Response
Manuscript ID: ijerph-1900052
International Journal of Environmental Research and Public Health
Increased risk of fractures and use of proton pump inhibitors in menopausal women: a systematic review and meta-analysis
October 1st, 2022
Dear Editor Dr. Agnieszka Matuszewska,
Please find enclosed the corrections of the manuscript “Increased risk of fractures and use of proton pump inhibitors in menopausal women: a systematic review and meta-analysis ” by Thuila Ferreira Maia, Bruna Gafo Camargo, Meire Ellen Pereira, Cláudia Sirlene de Oliveira, and Izonete Cristina Guiloski and the answers to the editor and reviewers. We still would like this manuscript can be considered for publication in International Journal of Environmental Research and Public Health. Best wishes, and thanks again for the opportunity to resubmit the revised version.
The English were revised and the title changed.
Reviewer 2
English language and style
( ) Extensive editing of English language and style required
(x) Moderate English changes required
( ) English language and style are fine/minor spell check required
( ) I don't feel qualified to judge about the English language and style
R: The English was revised and the title changed.
Comments and Suggestions for Authors
This systematic review and meta-analysis is an interesting paper. The paper discusses the association between PPI use and future fractures in post-menopausal women. The strength of this paper is that it is novel regarding this specific topic and it addresses an important issue. It follows most of PRISMA’s guidelines for systematic reviews and meta-analyses. However, several items of the PRISMA 2020 checklist are not addressed https://prisma-statement.org/ (methods: item 10b, 12-13c, 13f-15 of the checklist, results: 16b-18, 20a, 20c-22 of the checklist and several sections of the results, discussion and conclusion lack clarity and depth as specified in the comments.
R: All PRISMA topics cited have been revised and adjusted in the text.
Major and minor comments
Major
- Although the authors state to have used the PRISMA guidelines for a systematic review many items are missing (e.g. . Please elaborate on specific topics in the methods:
R: All PRISMA topics cited were inserted in the text.
- The data that is extracted from the original articles. Why was it not only the primary outcome?
- In addition to the main objective, other data were extracted from the articles, as they could interfere or act as confounding factors in the results.
- The date of data extraction should be added
R: This is described on line 100.
- Please elaborate on inclusion criteria for age, why do you choose such specific ages?
R: The menopause period does not have a pre-established age for each woman, so the age addressed in this review is an average of the ages of menopause in women reported in the studies used to develop the review.
- please state the elegibility criteria deriving from your PICO before the search strategy.
R: It was done.
- “Such as” line 101: is this an example of information that was extracted or a complete overview. Please give a complete overview as part of your methods according to the PRIMA guideline point 10a.
R: This has been adjusted according to the PRIMA guideline point 10a. Lines 104-112.
- The authors do not state their assumption strategy in case of missing data.
R: Articles that did not present data on the number of fractures, number of individuals who used PPIs, or at least relative risk were not included.
- paragraph 2.4: the threshold for high or low quality is missing
R: It was adjusted. lines 116-126.
- Information regarding the methods used in this meta-analysis is very limited. The metameter of interest (OR/HR etc.) is not named. The method for meta-analysis were not named.
R: More details have been added, topic 2.5, lines 132-142.
- Publication Bias was not assessed
R: We include the funnel plot in figure 2.
- possible sensitivity analysis was not stated.
R: We performed the test with all the studies that fit our question and when observing our funnel plot we saw that two studies were outside, when performing again without these studies we observed that the results did not change (as observed in figure 2), thus considering the reliable data.
- Results section
- A clear overview and description of each study (and citation) is lacking. Please describe the main outcome measures of the studies (RR,HR?) in the text. Not only in the table.
R: adjusted. topic 3.4, lines 189-233.
- Please, add references of included studies in the text of the results, as it is now unclear on which study you reflect (e.g line 129-131 but more examples could be given)
R: adjusted. Topic 2.5, lines 132-134.
- Line 163 -187: Please add effect sizes of included original research into the results
R: adjusted. Topic 3.4, lines 189-239.
- Line 160: Please give measures (RR 95%CI?) to interpret the numerical results in text
R: adjusted. Topic 3.4, lines 189-239.
- Please state in lines 123 and 124 why one of the studies was not included in the meta-analysis and which one.
R: adjusted. Topic 3.1, lines 150-152.
- “All studies had as an outcome the increase in fracture index related to the use of PPIs” It is unclear what is meant by the fracture index
R: A writing error has occurred, We meant the total number of fractures and not the fracture index.
- Discussion section
- Please review the references that are used. There are references are not suited for the purpose they are used for. E.g. line 45-48 states interference of PPI with calcium absorption and hypergastrenemia. However, they use a study on the relation between PPI and fractures as a reference (khalili etal). Khalili does not study the pharmacodynamics of PPI’s and their side-effects, they do mention the same mechanism in their introduction and reference this correctly. I would suggest referencing the original studies that do study the specifics on PPI use and possible mechanisms that could reduce BMD. Another example is the statement in line 33-34. “Their mechanism of action works by inhibiting the H+/K+-ATPase enzyme in gastric parietal cells, blocking acid secretion, and they are the most potent drugs available for this purpose”, while the main conclusion of the provided reference of Ali et al. is “Irrational prescribing of PPIs continues both in hospital and in general practice.”
R: Adjusted. Topic 4, lines 254-290.
- The exposure of PPI use in this meta-analysis may be of special importance.
Since the duration of ppi use was 12 months in 2 out of 5 included studies, a bit longer (up to 36 months) in only 2 studies and in 1 study duration was not provided, it is not feasible to draw firm conclusions on ‘long term’ ppi use which is done in the discussion and conclusion. The finding that fracture risk is increased already in the first year of ppi use may point at confounding since a biological explanation for a rapid deterioration of bone quality resulting in an increased fracture risk in the first year of ppi use is not really available. This should be discussed.
Same for ppi dose, which is not addressed in the meta-analysis
R: The topics were discussed. Topic 4, lines 267-270.
- The authors elaborate on the clinical characteristics that might add to or interfere with the association between PPI use and future fractures. But this is not an answer to the findings in the study related to the main research question. Also in the discussion the effect size they found in this meta-analysis is not compared to the existing body of literature.
R: Adjusted. Topic 4, lines 254-297.
- Although this is the first review on fracture risk in post-menopausal women. The article does not state clearly what the added benefit is in comparison to existing reviews and meta-analysis on the association between PPI use and Fractures.
Their aim is stated: “the objective of this systematic review was to assess whether there is a relationship between prolonged use of proton pump inhibitor drugs and fractures in menopausal women, as well as to assess whether the combination of these two factors can potentiate this risk.” The second part of the aim is not addressed in the results section. The answering of this question would require a comparison with risk measures from existing meta-analysis on the association between PPI use and fractures in the general population.
R: Unlike other published systematic reviews, we want to emphasize that menopausal women are more vulnerable to fractures when using some type of PPI. The second part of the objectives has been revised and modified (Lines 63-65)
The title of the manuscript is not in line with conclusion of the study and therefore incorrect: the word ‘causes’ suggests a direct relationship between ppi use and increased fracture risk. The main conclusion of this meta-analysis however is that it is quite likely that the association between ppi use and fracture risk as found in this study is distorted by counfounding.
R: The title was changed.
Minor
- Although I am not a native speaker, I have the impression that several sentences are not written in correct English.
R: The article was sent for correction by a specialized professional.
- The structure of sentences with many commas makes it sometimes difficult and sometimes incorrect read. For example: line 26-40, line 53-57. line 192-200
R: Adjusted.
- The rationale of some sentences is not clear:
line 135-138 Reported periods of menopause were post-menopause (this is not clear what you mean), aged 56 to 78.5 years (persons?), using some type of PPI and compared to the group that was not using (an PPI?), for a minimum period of 12 months (the use or the non-use) and that during the study had some type of fracture resulting from this use.
R: Adjusted. Topic 3.2, lines 167-172.
- Please note: The minimum required word count for reviews of this paper is 4000 words.
R: Adjusted. Words have been reduced to the maximum allowed.

Reviewer 3 Report
The manuscript by Thuila Ferreira da Maia et al claimed and concluded that the prolonged use of proton pump inhibitors by menopausal women can affect bone metabolism and cause fractures. However, the manuscript could be improved if the following comments and questions are addressed.
1. The figures are not clear. As the authors guidelines mentioned that Figures must be clear and readable, and we recommend a minimum resolution of 600 dpi.
2. In abstract, the authors mentioned that “The seven articles found in the databases, which met the eligibility criteria, cover menopausal women, aged between 56 and 78.5 years, using or not using some PPI for a minimum period of 12 months.” However, in the Results” Of these eight articles, one was excluded for being a narrative review article, two did not meet the eligibility criteria and the other five were selected for the development of the systematic review and four to the meta-analysis.” My main concern is related to the number.
3. I’m concerned if the participants (menopausal woman) in the manuscript have treated with anti-menopause or anti-osteoporosis drugs.
4. The manuscript should be thoroughly edited for grammar. It seems that the manuscript was written in parts, where some sections are with correct English, while other sections were difficult to read. Overall poorly edited manuscript including several typographical errors in scientific writing as well.
Author Response
Manuscript ID: ijerph-1900052
International Journal of Environmental Research and Public Health
Increased risk of fractures and use of proton pump inhibitors in menopausal women: a systematic review and meta-analysis
October 1st, 2022
Dear Editor Dr. Agnieszka Matuszewska,
Please find enclosed the corrections of the manuscript “Increased risk of fractures and use of proton pump inhibitors in menopausal women: a systematic review and meta-analysis ” by Thuila Ferreira Maia, Bruna Gafo Camargo, Meire Ellen Pereira, Cláudia Sirlene de Oliveira, and Izonete Cristina Guiloski and the answers to the editor and reviewers. We still would like this manuscript can be considered for publication in International Journal of Environmental Research and Public Health. Best wishes, and thanks again for the opportunity to resubmit the revised version.
The English were revised and the title changed.
Reviewer 3
English language and style
( ) Extensive editing of English language and style required
(x) Moderate English changes required
( ) English language and style are fine/minor spell check required
( ) I don't feel qualified to judge about the English language and style
R: The English was revised and the title changed.
Comments and Suggestions for Authors
The manuscript by Thuila Ferreira da Maia et al claimed and concluded that the prolonged use of proton pump inhibitors by menopausal women can affect bone metabolism and cause fractures. However, the manuscript could be improved if the following comments and questions are addressed.
- The figures are not clear. As the authors guidelines mentioned that Figures must be clear and readable, and we recommend a minimum resolution of 600 dpi.
R: Figures have been adjusted for better viewing.
- In abstract, the authors mentioned that “The seven articles found in the databases, which met the eligibility criteria, cover menopausal women, aged between 56 and 78.5 years, using or not using some PPI for a minimum period of 12 months.”However, in the Results” Of these eight articles, one was excluded for being a narrative review article, two did not meet the eligibility criteria and the other five were selected for the development of the systematic review and four to the meta-analysis.” My main concern is related to the number.
R: Adjusted according to the flowchart data. Line 22-24.
- I’m concerned if the participants (menopausal woman) in the manuscripthave treated with anti-menopause or anti-osteoporosis drugs.
R: According to table S2, the authors of the studies report that the evaluated women could use both hormonal therapy and anti-osteoporosis medications.
- The manuscript should be thoroughly edited for grammar. It seems that the manuscript was written in parts, where some sections are with correct English, while other sections were difficult to read. Overall poorly edited manuscript including several typographical errors in scientific writing as well.
R: Adjusted. The article was sent for correction by a specialized professional.

Round 2
Reviewer 2 Report
The paper has been improved significantly compared to the first version. I have no further comments.
Reviewer 3 Report
The authors have addressed my concerns. I find the language still needs some correction.